# Pregnancy Outcomes of Freeze-All versus Fresh Embryo Transfer in Women with Adenomyosis: A Retrospective Study

**DOI:** 10.3390/jcm12051740

**Published:** 2023-02-21

**Authors:** Qiaoli Zhang, Qiaoyuan Chen, Tianhe Li, Zhaoxia Jia, Xiaomeng Bu, Yanjun Liu, Shuyu Wang, Ying Liu

**Affiliations:** 1Department of Human Reproductive Medicine, Beijing Obstetrics and Gynecology Hospital, Capital Medical University, Beijing Maternal and Child Health Care Hospital, 251 Yaojiayuan Road, Beijing 100026, China; 2Central Laboratory, Beijing Obstetrics and Gynecology Hospital, Capital Medical University, Beijing Maternal and Child Health Care Hospital, 251 Yaojiayuan Road, Beijing 100026, China; 3Department of Information and Statistics, Beijing Obstetrics and Gynecology Hospital, Capital Medical University, Beijing Maternal and Child Health Care Hospital, 251 Yaojiayuan Road, Beijing 100026, China

**Keywords:** adenomyosis, infertility, in vitro fertilization, freeze-all, fresh embryo transfer, pregnancy outcome

## Abstract

Adenomyosis has been associated with adverse fertility and pregnancy outcomes, and its impact on the outcomes of in vitro fertilization (IVF) has received much attention. It is controversial whether the freeze-all strategy is better than fresh embryo transfer (ET) in women with adenomyosis. Women with adenomyosis were enrolled in this retrospective study from January 2018 to December 2021 and were divided into two groups: freeze-all (*n* = 98) and fresh ET (*n* = 91). Data analysis showed that freeze-all ET was associated with a lower rate of premature rupture of membranes (PROM) compared with fresh ET (1.0% vs. 6.6%, *p* = 0.042; adjusted OR 0.17 (0.01–2.50), *p* = 0.194). Freeze-all ET also had a lower risk of low birth weight compared with fresh ET (1.1% vs. 7.0%, *p* = 0.049; adjusted OR 0.54 (0.04–7.47), *p* = 0.642). There was a nonsignificant trend toward a lower miscarriage rate in freeze-all ET (8.9% vs. 11.6%; *p* = 0.549). The live birth rate was comparable in the two groups (19.1% vs. 27.1%; *p* = 0.212). The freeze-all ET strategy does not improve pregnancy outcomes for all patients with adenomyosis and may be more appropriate for certain patients. Further large-scale prospective studies are needed to confirm this result.

## 1. Introduction

Adenomyosis is a common gynecological disease in women of childbearing age [1] and has been vaguely described as a benign condition of endometrial tissue that involves gland and stromal invasion into the myometrium, thus leading to an enlarged uterus [2]. The different clinical manifestations of adenomyosis are as follows: 40–60% of patients have excessive menstruation, 15–30% of patients have dysmenorrhea, and 30% of patients have no obvious symptoms [3]. Gynecological transvaginal ultrasound examination found that the uterus was enlarged in women with adenomyosis. The inhomogeneity of the original base echo and endometrial line deviation are important factors for judging adenomyosis [4]. Women with adenomyosis suffer from sub- and infertility with low fecundity [5]. The prevalence of adenomyosis in infertile women has been reported to be approximately 22% in women aged less than 40, and approximately 24.4% in women aged 40 years and above [6].

Various mechanisms have been proposed to explain the potential negative impact of adenomyosis on fertility. In women with adenomyosis, the normal architecture of the “archimyometrium” (junctional zone myometrium) was destroyed owing to invagination of the endometrial glands and stroma, thus leading to abnormal uterine contractility, such as impairment of the uterine system of sperm transport secondary to the alteration of the normal myometrial structure, and the presence of uterine dysperistalsis, including altered endometrial function and receptivity [7], with a consequent reduction in the probability of embryo implantation [8]. With intrauterine abnormalities and anatomical distortion of the uterine cavity, women with adenomyosis were found to have a lower clinical pregnancy rate and a higher miscarriage rate [9]. Adenomyosis can trigger endometrial inflammation and lead to histological modifications that could favor abnormal decidualization, and potentially defective placentation [10].

There has been discussion in recent years of a freeze-all strategy, which involves freezing all embryos, then thawing and transferring embryos into a more physiological environment in subsequent appropriate cycles. By adopting this strategy, the potential deleterious effects of ovarian stimulation on the endometrium could be avoided, and better results would be obtained. To date, no studies have elucidated whether the freeze-all strategy is beneficial in improving pregnancy outcomes in patients with adenomyosis. Therefore, the purpose of this study was to assess whether the freeze-all strategy is associated with improvements in assisted reproduction technology (ART) outcomes compared with fresh embryo transfer in in vitro fertilization (IVF)/intracytoplasmic sperm injection (ICSI), and to identify therapeutic strategies for improving fertility in adenomyosis patients.

## 2. Materials and Methods

### 2.1. Study Design and Subject Recruitment

We conducted a retrospective cohort study at the Department of Human Reproductive Medicine, Beijing Obstetrics and Gynecology Hospital, Capital Medical University, Beijing. The data were analyzed from 189 infertile women with adenomyosis who underwent IVF or ICSI between January 2018 and December 2021. Data from couples who underwent IVF/ICSI included the tubal factor, ovulatory dysfunction, endometriosis, male factor, and diminished ovarian reserve (DOR). The criteria for the women involved in this study included those: (1) diagnosed with adenomyosis by ultrasound; (2) scheduled for either a fresh embryo transfer (ET) or a freeze-all strategy in both IVF and ICSI cycles; and (3) with follow up available up to the end of pregnancy. Exclusion criteria included: (1) patients with all embryos frozen without transfer and no transferable embryos; (2) diagnosis of intrauterine disease, including endometrial polyps or submucosal myomas, uterine malformations (unicornuate, bicornuate, or septate), and intrauterine adhesion, as determined by ultrasound or hysteroscopy; (3) patients lost to follow up; (4) patients with polycystic ovary syndrome (PCOS); and (5) patients with cancer history. Finally, 189 cycles were divided into a fresh group (*n* = 91) and a freeze-all group (*n* = 98). Figure 1 shows the flow chart of this study (Figure 1).

The sonographic diagnosis criteria of adenomyosis by ultrasound were as follows: ill-defined hypo-echogenic area, globular enlarged uterus, myometrial cysts, poor definition of the endometrial myometrial junction, asymmetry of the anteroposterior myometrium, and heterogeneous myometrial echogenicity [11,12]. Calculation of uterine volume was based on the following geometric formula: long diameter × width diameter × anteroposterior diameter × π/6 [13,14].

### 2.2. Protocols for Controlled Ovarian Stimulation

The following different controlled ovarian stimulation (COS) protocols were individualized based on the woman’s age, ovarian reserve and response, as well as menstrual cycle: a long gonadotrophin-releasing hormone (GnRH) agonist protocol; an ultralong GnRH agonist protocol; a short GnRH agonist protocol; a GnRH antagonist protocol; and a progestin primed ovarian stimulation (PPOS) protocol [15,16].

Follicular development was monitored by transvaginal ultrasound (TVS), and serum E_2_, P, and LH levels were measured by chemiluminescence. When echography revealed at least three follicles >18 mm in diameter, 250 μg recombinant human chorionic gonadotropin (hCG) (Ovidrel^®^, Merck, Modugno, Italy) or 0.2 mg GnRH agonist (Decapeptyl^®^, 0.1 mg, Ferring, Kiel, Germany) was administered subcutaneously. Transvaginal ultrasound-guided oocyte retrieval was then scheduled for 36 h after the trigger. The retrieved oocytes were inseminated by means of IVF or ICSI according to the quality of the spermatozoa.

The choice of the freeze-all embryo transfer for patients with high ovarian hyperstimulation syndrome risk, severe adenomyosis, elevated progesterone on hCG administration day, thin endometrium, uterine effusion, hydrosalpingeal fluid, and uncontrolled medical disease, was a joint decision made by the patient and the doctor. Fresh ET was applied for patients with mild adenomyosis, normal ovarian reserve, and young age.

Depending on the number and quality of embryos, the severity of adenomyosis, the age of the patient, as well as the patient’s wish and the doctor’s preference, single or double embryos were transferred.

### 2.3. Fresh Embryo Transfer

After oocyte retrieval, 1–2 cleavage embryos, including day 2, day 3 (according to the Peter grading system [17]), day 4 morula embryos [18] or day 5–6 blastocysts (according to the Gardner grading system [19]), were selected to be transferred into the uterine cavity. Luteal support was initiated with oral progesterone and vaginal progesterone on the day of oocyte retrieval. Luteal support was continued until 10 weeks of gestation if pregnancy was achieved.

### 2.4. Frozen Embryo Transfer

The endometrial preparation was performed under the natural cycle, the ovulation cycle, or the hormone replacement therapy (HRT) cycle with exogenous estrogen, according to the patient’s condition [20]. The HRT cycle can be initiated with or without GnRH agonist pretreatment [21]. Estradiol valerate of 6 mg (Progynova, DELPHARM Lille S.A.S., Lille, France) was orally administered about 2 weeks for endometrial preparation until the initial pregnancy test [22]. The patients were given oral progesterone and intravaginal progesterone on the day of ovulation, or when the endometrial thickness was ≥7 mm. One to two cleavage-stage embryos or morula embryos or blastocysts were thawed for transfer. Similarly, luteal support was continued after pregnancy was achieved, until 10 weeks of gestation.

If the patients with adenomyosis had undergone several frozen embryo transfer (FET) cycles, only their first cycle was considered to exclude bias in the statistical analysis.

### 2.5. Study Outcomes

The primary outcomes were the clinical pregnancy rate and live birth rate (LBR) per ET cycle. The secondary outcomes were: rates of biochemical pregnancy, ongoing pregnancy, miscarriage, preterm birth, singleton live birth, premature rupture of membranes (PROM), hypertensive disorders of pregnancy, and low-birth-weight infant per ET cycle.

A biochemical pregnancy was defined as a concentration of >5 IU/L serum β-hCG on day 12–14 after cleavage-stage ET, or day 10–12 after morula and blastocyst transfer. In biochemical pregnancy cases, ultrasound examination was performed 35 days after transfer. A clinical pregnancy was defined as the presence of at least one gestational sac, including ectopic pregnancy. An ongoing pregnancy was defined as ultrasound-confirmed evidence of a gestational sac with fetal heart motion at 12 weeks [23]. A miscarriage was defined as the loss of a clinical pregnancy before 28 weeks of gestation. A live birth was defined as any birth event in which at least one baby was born alive. Preterm birth was defined as a birth that occurred 28 - < 37 weeks of gestational age. A low-birth-weight infant was defined as a birth weight of < 2500 g [24].

### 2.6. Statistical Analysis

Statistical analysis was performed using SPSS (Version 19, SPSS Inc., Chicago, IL, USA). The results of the study are expressed as the means ± standard deviations (SD), numbers (percentages), or medians (first quartiles–third quartiles). For continuous variables, the Kolmogorov–Smirnov test and Q-Q plots of normality were performed to choose the appropriate statistical test. Chi-square tests or Student’s *t* tests were performed to evaluate the significant differences between the variables. The association between the type of embryo transfer (freeze-all ET or fresh ET) and pregnancy outcomes was evaluated by binary logistic regression analysis while adjusting for potential confounders. The potential confounders were determined by statistical significance, including anti-Müllerian hormone (AMH), basal follicle stimulating hormone (FSH), duration of ovarian stimulation in days, total dose of gonadotropin (Gn), luteinizing hormone (LH) levels and endometrial thickness on the day of hCG injection, and type of embryo transferred. The results of logistic regression were displayed as odds ratio (OR) and 95% confidence interval (CI). A value of *p* < 0.05 was considered statistically significant.

## 3. Results

### 3.1. Baseline Characteristics

The patients’ baseline characteristics and clinical features are shown in Table 1. There were no significant differences between the two groups with regard to age, body mass index (BMI), duration of infertility, basal LH levels, basal estradiol (E_2_) levels, uterine volume, or type of infertility (all *p* > 0.05).

In addition, compared to the fresh ET group, AMH levels were lower in the freeze-all group, but basal FSH levels were higher (all *p* < 0.05).

### 3.2. Ovarian Stimulation and Transfer Characteristics Outcomes

Compared to the fresh ET group, the LH levels on the day of hCG were higher (*p* < 0.05) and the duration of ovarian stimulation in days, the total amount of Gn administered, and the endometrial thickness on the day of hCG injection were lower in the freeze-all group (all *p* < 0.05). The type of embryo transferred between the two groups was significantly different (*p* < 0.05). The remaining subjects’ baseline characteristics and treatment characteristics were similar between the two groups (all *p* > 0.05) (Table 2).

### 3.3. Pregnancy Outcomes and Obstetric Complications

The pregnancy outcomes and obstetric complications are listed in Table 3. The PROM rate per cycle was significantly lower in the freeze-all group than that of the fresh group (1.0% vs. 6.6%; *p* = 0.042). The freeze-all group also had a lower risk of low-birth-weight infants per cycle than the fresh ET group (1.1% vs. 7.0%; *p* = 0.049). Despite a trend toward a lower miscarriage rate and preterm birth rate after freeze-all, the difference did not reach statistical significance (Table 3).

In addition, the rates of biochemical pregnancy, clinical pregnancy, ongoing pregnancy, live birth, singleton live birth, and hypertensive disorders of pregnancy per cycle were not significantly different between the two groups (Table 3).

To identify whether the type of ET (freeze-all or fresh transfer) was a risk factor for pregnancy outcomes after IVF/ICSI, relevant subject characteristics were used as potential confounders. There were significant differences between the two groups in levels of AMH, basal FSH, duration of ovarian stimulation in days, total dose of Gn, LH levels and endometrial thickness on the day of hCG injection, and type of embryo transferred (Table 1 and Table 2). The logistic regression results showed no significant difference in PROM risk between the freeze-all and fresh groups (the adjusted OR for the freeze-all group was 0.17 (0.01–2.50), *p* = 0.194)) after adjustment for potential confounders. Similarly, there was no significant difference in low-birth-weight risk between the two groups (adjusted OR 0.54 (0.04–7.47), *p* = 0.642) (Table 4).

## 4. Discussion

Adenomyosis could impair fecundity. Women with adenomyosis have disturbed uterine peristalsis and uterotubal transport abnormalities. Adenomyoma distorts the uterine cavity, and myometrial structure and function are severely impaired. Endometrial function and receptivity are altered by adenomyoma, with one of the effects being abnormal endometrial metabolism [25]. Compared with women without adenomyosis, women with adenomyosis have poor reproductive outcomes when undergoing IVF/ICSI [25]. This retrospective cohort study aimed to assess whether the freeze-all strategy results in better outcomes than fresh transfer, and it was found that the freeze-all strategy does not improve the pregnancy outcomes of women with adenomyosis in the first transfer cycle.

A meta-analysis showed that FET led to significantly higher rates of implantation, ongoing pregnancy and clinical pregnancy than fresh ET. The better embryo-endometrium synchrony could be the explanatory factor [26]; the embryos avoid suffering from the supraphysiologic hormonal levels experienced during controlled ovarian hyperstimulation [27]. Freeze-all strategies may be advantageous when collecting large numbers of oocytes, signaling an association between higher ovarian stimulation and impaired endometrial receptivity; however, they are not advantageous when the mean number of oocytes collected is less than 15 [28]. In our study, compared to the fresh ET group, AMH levels were lower (1.97 ± 2.23 ng/mL vs. 2.89 ± 2.53 ng/mL, *p* < 0.05) and basal serum FSH levels were higher (8.66 ± 4.63 mIU/mL vs. 7.02 ± 2.43 mIU/mL, *p* < 0.05) in the freeze-all group. This information indicates that patients with adenomyosis had lower ovarian reserve should prefer to be given a freeze-all strategy; additionally for those with larger uterine volumes. Since embryos accumulate, the uterus is pretreated before FET, and the embryos are transferred at the right time.

As a consequence of destroying the normal architecture of the myometrium [29], adenomyosis may decrease implantation [2,30] and the clinical pregnancy rate in IVF treatment [2]. Embryo implantation failure was found to be high when a pelvic magnetic resonance imaging (MRI) scan showed that the average thickened junctional zone was greater than 7 mm after IVF [31]. Women with adenomyosis (40.5%) had a significant reduction in clinical pregnancy compared to those without adenomyosis (49.8%) [32]. Patients with adenomyosis trended more toward an increased clinical pregnancy rate following FET with GnRH-a pretreatment, compared with fresh ET, but the difference was nonsignificant (39.5% vs. 30.5%, *p* > 0.05) [22]. A meta-analysis showed that women with adenomyosis have a lower clinical pregnancy rate (OR 0.66, 95% CI 0.48–0.90) and ongoing pregnancy rate (OR 0.43, 95% CI 0.21–0.88), than those without adenomyosis [9]. Our results also showed that the women with adenomyosis had lower clinical pregnancy rate and ongoing pregnancy rate, and there were no significant differences in the clinical pregnancy rate (33.3% vs. 46.1%, *p* > 0.05) or ongoing pregnancy rate (21.3% vs. 33.7%, *p* > 0.05) between the freeze-all group and the fresh ET group in women with adenomyosis. The clinical pregnancy rates (56.4 vs. 31.5%, *p* < 0.05) were significantly reduced in a group of patients with a myometrial thickness of more than 2.5 cm compared with that <2.0 cm [33]. The estimated probability of clinical pregnancy was inversely correlated with adenomyosis severity scores, which decreased from 42.7% (95% CI 37.1–48.3) for women with no adenomyosis features to 22.9% (95% CI 13.4–32.6); and 13.0% (95% CI 2.2–23.9) for those with four or all seven features [34]. These data should inspire research based on the severity of the characteristics of adenomyosis in the future.

In a previous study, the rate of miscarriage was significantly higher in the adenomyosis group than in the control group [35]. The presence of abnormally high levels of free radicals in the uterine milieu of women with adenomyosis could create an unfavorable environment for embryo development, and consequently enhance the risk of early miscarriage [10]. The meta-analysis showed that women with adenomyosis had a higher miscarriage rate (OR 2.11, 95% CI 1.33–3.33) than those without adenomyosis [9], and another meta-study showed similar results [2]. The patients with adenomyosis accompanied by a larger uterine volume (>98.81 cm^3^) prior to FET suffered an increased risk of miscarriage (51.28% vs. 16.28%, *p* = 0.001) compared with those with a smaller uterine volume (≤98.81 cm^3^) [36]. In another study, the miscarriage rates were significantly higher in patients with a myometrial thickness of more than 2.5 cm [33]. These data indicated that the risk of miscarriage in women with adenomyosis is related to the uterine volume and the severity of the disease. The miscarriage rate (8.9% vs. 11.6%, *p* > 0.05) in our study did not differ significantly between the freeze-all group and the fresh ET group in women with adenomyosis. Follow-up studies should analyze miscarriage rates in combination with the characteristics of the uterus.

In another previous study, women with adenomyosis had a lower LBR (OR 0.59, 95% CI 0.37–0.92, *p* = 0.02) than those without adenomyosis [9]. A meta-analysis showed that the presence of adenomyosis was associated with a 41% decrease in LBR after IVF (OR 0.59, 95% CI 0.42–0.82) [2]. The same results were found in the matched control group of a separate study [37]. Women with adenomyosis have a significant reduction in delivery rates (26.8% vs. 37.1% in women with and without adenomyosis) [32] and term pregnancy rate [35]. A systematic review and meta-analysis indicated higher LBRs by FET than by fresh ET in hyper-responders (RR 1.16, 95% CI: 1.05–1.28). However, there was no difference in LBR among normal responders (RR 1.03; 95% CI: 0.91–1.17) [38]. Women with adenomyosis who have normal ovarian reserves may benefit from a fresh ET with or without 3 months of pretreatment with GnRH-analog [39]. One study with 158 adenomyosis patients indicated that uterine volume before FET was negatively correlated with live birth [36]. Our results indicated that there was no higher LBR in the freeze-all group compared with the fresh group in women with adenomyosis (19.1% vs. 27.1%, *p* > 0.05).

Adenomyosis has been linked to several obstetric complications, such as preeclampsia, PROM, preterm birth, small-for-gestational-age/fetal growth restriction, and placental abruption [40]. In a nested case-control study, women with adenomyosis had a higher risk of spontaneous preterm delivery (OR 1.84, 95% CI 1.32–4.31) and PROM (OR 1.98, 95% CI 1.39–3.15) than those without adenomyosis [41]. A Japanese retrospective case-control study clarified that an increased risk of preterm birth was attributed to adenomyosis by matched analysis adjusted for age and ART therapy (24.4% vs. 9.3%, OR 3.1, 95% CI 1.2–7.2) [42]. Furthermore, another study demonstrated higher rates of extreme preterm delivery (41.7% vs. 12.5%, OR 4.3, 95% CI 1.0–18.4), PROM (41.7% vs. 12.5%, OR 5.5, 95% CI 1.7–17.7), and small-for-gestational-age (33.3% vs. 10.4%, OR 4.3, 95% CI 1.8–10.3) compared with the control group [43]. The risks of low birth weight of <2500 g (OR 1.83, 95% CI 1.36–2.45) and low birth weight of <1500 g (OR 2.39, 95% CI 1.20–4.77) were higher in women with adenomyosis [44]. Compared with fresh ET, FET was associated with a lower risk of preterm birth (7.0% versus 17.5%; *p* = 0.010) among women with adenomyosis [45]. Our study showed no difference in the preterm birth rate between the two groups, which may be related to the inclusion of only the first FET cycle in the freeze-all group in our study. The adenomyosis of the uterus can be classified into focal and diffuse types. The risk of preterm birth (OR 5.24, 95% CI 2.15–12.8) and PROM (OR 5.56, 95% CI, 1.42–21.7) in women with diffuse-type adenomyosis was significantly increased compared with that in women without adenomyosis, but the risk of preterm birth in the focal-type group was not found to be higher than that in women without adenomyosis [46]. Our results showed that there was a tendency to reduce the risk for PROM and low birth weight in women who underwent freeze-all ET, compared with those who underwent fresh ET. However, the adjusted ORs for the fresh transfer, after adjustment for potential confounders, have no statistical differences.

A limitation of this study is that it is a retrospective study with a small sample size. Also, different endometrial preparation protocols were employed for FET. Another possible limitation may have resulted from the misdiagnosis of adenomyosis, which could induce potential bias. Adenomyosis affects pregnancy outcomes differently depending on the severity of uterine involvement and subtype. The severity of adenomyosis was not taken into account in this study. Despite these limitations, the present study has several strengths. The impact of the freeze-all strategy on the pregnancy outcomes of women with adenomyosis has been emphatically demonstrated. Additionally, this study provided a comprehensive analysis, and included pregnancy outcomes and obstetric complications. Therefore, additional well-designed RCTs are needed to evaluate the freeze-all strategy for women with adenomyosis. Further research that considers the type or severity of adenomyosis is needed.

In conclusion, the freeze-all strategy has been found to benefit specific populations, but our results indicate that the freeze-all strategy is controversial and does not improve pregnancy outcomes when compared with fresh ET cycles after IVF/ICSI among women with adenomyosis. In contrast, the freeze-all strategy may require additional embryo manipulation; increase in the cost of treatment and time to live birth for patients; and increase in the workload of clinicians. The present data suggest that the freeze-all strategy should be individualized in line with precision medicine for adenomyosis patients with potential benefits, which is the focus of our future research.

## Figures and Tables

**Figure 1 jcm-12-01740-f001:**
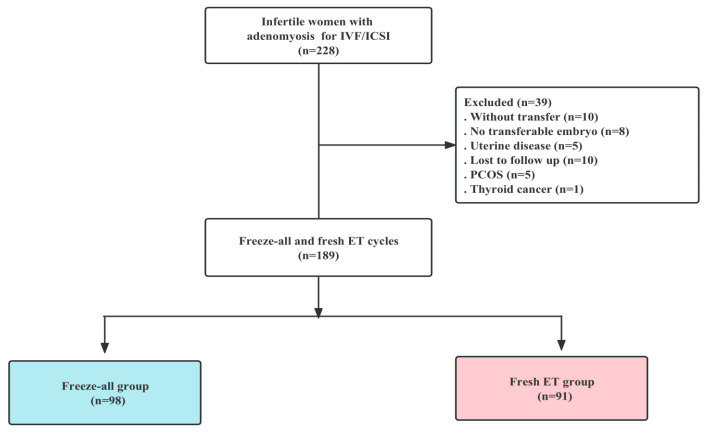
Flow chart of the cohort screening in the study.

**Table 1 jcm-12-01740-t001:** Baseline characteristics and clinical features of patients in the two groups.

Variables	Freeze-All Group (*n* = 98)	Fresh Group (*n* = 91)	*p* Value
Age (y)	35.05 ± 4.67	34.67 ± 3.91	0.546
BMI (kg/m^2^)	22.87 ± 3.75	23.81 ± 3.97	0.099
Duration of infertility (y)	3.74 ± 2.34	3.29 ± 2.12	0.171
AMH (ng/mL)	1.97 ± 2.23	2.89 ± 2.53	0.014
Basal FSH (mIU/mL)	8.66 ± 4.63	7.02 ± 2.43	0.004
Basal LH (mIU/mL)	4.16 ± 3.26	3.99 ± 2.37	0.714
Basal E_2_ (pg/mL)	47.17 ± 23.66	47.11 ± 35.33	0.990
Uterine volume (cm^3^)	101.13 ± 68.49	86.84 ± 48.52	0.168
Type of infertility, *n* (%)			0.701
Primary infertility	48 (49.0)	42 (46.2)	
Secondary infertility	50 (51.0)	49 (53.8)	

Note: Values represent means ± standard deviation (SD) or number (percentage).

**Table 2 jcm-12-01740-t002:** Ovarian stimulation, embryology and transfer data between freeze-all and fresh groups.

Variables	Freeze-All Group (*n* = 98)	Fresh Group (*n* = 91)	*p* Value
Antral follicle count (AFC)	7.53 ± 4.94	8.86 ± 4.35	0.053
E_2_ on the day of trigger (pg/mL)	2477.15 ± 1873.64	2140.75 ± 1355.09	0.171
LH on the day of trigger (mIU/mL)	2.62 ± 2.37	1.59 ± 2.61	0.007
Gn dose (IU)	2238.25 ± 861.54	2792.58 ± 845.88	0.000
Stimulation length (d)	9.07 ± 2.46	10.89 ± 2.24	0.000
Endometrial thickness on the day of trigger (mm)	9.36 ± 2.46	10.81 ± 2.45	0.000
Method of fertilization, *n* (%)			0.247
IVF	91 (92.9)	80 (87.9)	
ICSI	7 (7.1)	11 (12.1)	
No. of oocytes retrieved	5 (2.0, 10.0)	7 (4.0, 11.0)	0.100
No. of mature oocytes	4 (2.0, 8.3)	5 (3.0, 10.0)	0.096
No. of oocytes fertilized	4 (2.0, 7.8)	5 (3.0, 8.0)	0.112
No. of 2 PN oocytes	3 (2.0, 6.0)	3 (2.0, 6.0)	0.752
No. of viable embryos	2 (1.0, 4.0)	2.5 (2.0, 4.0)	0.395
No. of high-quality embryos	1 (0, 2.0)	1 (0, 2.0)	0.914
No. of embryos transferred	1.72 ± 0.45	1.75 ± 0.44	0.730
No. of high-quality embryos transferred	1.05 ± 0.82	0.96 ± 0.81	0.766
No. of embryos transferred, *n* (%)			0.648
Single embryo transfer	28 (28.3)	23 (25.3)	
Double embryo transfer	70 (71.7)	68 (74.7)	
Type of embryo transferred, *n* (%)			0.015
D2 cleavage stage	33 (33.7)	16 (17.8)	
D3 cleavage stage	52 (53.3)	62 (67.8)	
D4 morula stage	0 (0)	4 (4.4)	
Blastocyst	13 (13.0)	9 (10.0)	

**Table 3 jcm-12-01740-t003:** Pregnancy outcomes and obstetric complications between freeze-all and fresh groups.

Variables	Freeze-All Group (*n* = 98)	Fresh Group (*n* = 91)	*p* Value
Biochemical pregnancy, *n* (%)	45 (45.8)	51 (56.0)	0.163
Clinical pregnancy, *n* (%)	33 (33.3)	42 (46.1)	0.082
Ongoing pregnancy, *n* (%)	21 (21.3)	31 (33.7)	0.067
Miscarriage, *n* (%)	9 (8.9)	11 (11.6)	0.549
Live birth, *n* (%)	19 (19.1)	25 (27.1)	0.212
Preterm birth, *n* (%)	2 (2.2)	6 (7.0)	0.134
Singleton live birth, *n* (%)	18 (18.4)	23 (25.3)	0.223
PROM, *n* (%)	1 (1.0)	6 (6.6)	0.042
Hypertensive disorders of pregnancy, *n* (%)	5 (5.6)	3 (3.5)	0.500
Low-birth-weight infant, *n* (%)	1 (1.1)	6 (7.0)	0.049

**Table 4 jcm-12-01740-t004:** Adjusted OR of pregnancy outcomes and obstetric complications between freeze-all and fresh groups.

Variables	Adjusted OR (95% CI)	*p* Value
Biochemical pregnancy	0.79 (0.35–1.81)	0.589
Clinical pregnancy	0.76 (0.32–1.81)	0.541
Ongoing pregnancy	1.34 (0.46–3.94)	0.594
Miscarriage	0.61 (0.13–2.79)	0.520
Live birth	1.55 (0.52–4.64)	0.432
Preterm birth	0.89 (0.09-8.46)	0.922
Singleton live birth	1.26 (0.40–3.95)	0.693
PROM	0.17 (0.01–2.50)	0.194
Hypertensive disorders of pregnancy	3.26 (0.43–24.62)	0.253
Low-birth-weight infant	0.54 (0.04–7.47)	0.642

Note: The fresh embryo transfer as reference.

## Data Availability

The datasets in this study are available from the corresponding authors.

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
