# Peer review of "Pregnancy Outcomes of Freeze-All versus Fresh Embryo Transfer in Women with Adenomyosis: A Retrospective Study"

_jcm, 2023, doi:10.3390/jcm12051740_

Round 1
Reviewer 1 Report
This is a well-designed and executed study investigating an interesting topic. Study methodology, data analysis and interpretation are of good quality and conclusions drawn are supported by the findings of the study. According to this reviewer’s opinion this study merits publication. However, some major revisions are needed prior to publication based on the following comments.
1. Considering the retrospective nature of the study and in order to minimize bias, authors should clearly state exclusion and inclusion criteria employed for patient stratification in the study groups. To elaborate on that, authors should clearly state the criteria used to decide whether a patient with adenomyosis was eligible to receive FET or fresh cycle management. Moreover, the number of the patients excluded from the analysis should be highlighted, indicating the reasons for exclusion. The manuscript will be benefit by presenting a detailed flow-chart, indicating the selection process.
2. Authors acknowledged the presence of several confounders and thus proceeded to adjustment employing logistic regression analysis. However, the type of the respective confounders, for which adjustment was performed, should be highlighted in the material and methods section.
3. Authors should revise the sections 2.3 and 2.4 to avoid confusion of the readership. In Table 2 is clearly presented that ETs in both groups included embryos in all stages of preimplantation development, namely Day 2 and Day 3 cleavage stage embryos, morula stage embryos (Day 4) and Blastocyst stage embryos. In the material and methods section this should be clearly stated. Day 4 embryos are not considered to be in the cleavage stage but instead as in the morula stage. Moreover, authors noted that blastocyst stage embryos were morphologically evaluated employing the Gardner’s grading system. However, information regarding embryo selection process for embryo in the cleavage and morula stages is missing and should be clearly presented. In addition, authors should clearly state decision-making process about single or double ET.
4. Ethics Board approval is commonly needed for studies evaluating retrospective data. Authors should provide the protocol number of the approval. If no approval has granted, authors should clearly state the reason.
It would be my pleasure to see the revised version of the manuscript.
Author Response
- Considering the retrospective nature of the study and in order to minimize bias, authors should clearly state exclusion and inclusion criteria employed for patient stratification in the study groups. To elaborate on that, authors should clearly state the criteria used to decide whether a patient with adenomyosis was eligible to receive FET or fresh cycle management. Moreover, the number of the patients excluded from the analysis should be highlighted, indicating the reasons for exclusion. The manuscript will be benefit by presenting a detailed flow-chart, indicating the selection process.
Response: Thanks for your comments and advice throughout the revision process.
We appreciate the reviewer’s suggestion to supplement the critical information.
Couples for IVF/ICSI included the tubal factor, ovulatory dysfunction, endometriosis, male factor, and diminished ovarian reserve (DOR).
The following women were included: (1) diagnosed with adenomyosis by ultrasound; (2) scheduled for either a fresh embryo transfer (ET) or a freeze-all strategy in both IVF and ICSI cycles; (3) follow up available up to the end of pregnancy. Exclusion criteria included: (1) patients with all embryos were frozen without transfer and no transferable embryos; (2) diagnosis of intrauterine disease including endometrial polyps or submucosal myomas, uterine malformations (unicornuate, bicornuate, or septate), and intrauterine adhesion, as determined by ultrasound or hysteroscopy; (3) patients lost follow up; (4) patients with polycystic ovary syndrome (PCOS); (5) patients with cancer history.
The choice of the freeze-all embryo for patients with high ovarian hyperstimulation syndrome risk, severe adenomyosis, elevated progesterone on hCG administration day, thin endometrium, uterine effusion, hydrosalpingeal fluid, uncontrolled medical disease, and a joint decision by the patient and the doctor. The application of fresh ET for patients with mild adenomyosis, normal ovarian reserve, and young.
We presented a detailed flow-chart, indicating the selection process.
Figure 1. Flow chart of cohort screening in the study.
- Authors acknowledged the presence of several confounders and thus proceeded to adjustment employing logistic regression analysis. However, the type of the respective confounders, for which adjustment was performed, should be highlighted in the material and methods section.
Response: Thank you for focusing our attention here; we have added more details of our method in the revised manuscript (In the Statistical Analysis section).
The association between the type of embryo transfer (freeze-all ET or fresh ET) and pregnancy outcomes was evaluated by binary logistic regression analysis while adjusting for potential confounders. The potential confounders were determined by statistical significance, including anti-Müllerian hormone (AMH), basal follicle stimulating hormone (FSH), duration of ovarian stimulation in days, total dose of gonadotropin (Gn), luteinizing hormone (LH) levels and endometrial thickness on the day of hCG injection, and type of embryo transferred. The results of logistic regression were displayed as odds ratio (OR) and 95% confidence interval (CI).
- Authors should revise the sections 2.3 and 2.4 to avoid confusion of the readership. In Table 2 is clearly presented that ETs in both groups included embryos in all stages of preimplantation development, namely Day 2 and Day 3 cleavage stage embryos, morula stage embryos (Day 4) and Blastocyst stage embryos. In the material and methods section this should be clearly stated. Day 4 embryos are not considered to be in the cleavage stage but instead as in the morula stage. Moreover, authors noted that blastocyst stage embryos were morphologically evaluated employing the Gardner’s grading system. However, information regarding embryo selection process for embryo in the cleavage and morula stages is missing and should be clearly presented. In addition, authors should clearly state decision-making process about single or double ET.
Response: Thank you for helping to improve the manuscript.
The references of the embryo at the cleavage stage were added: Brinsden PR.A textbook of in vitro fertilization and assisted reproduction[M].New York: The Parthenon Publishing Groupe Inc,1999,196.
The sections 2.3, 2.4, 2.5, and table 2, have been revised to morula stage embryos (Day 4). (the added reference: Tao J, Craig RH, Johnson M, Williams B, Lewis W, White J, Buehler N. Cryopreservation of human embryos at the morula stage and outcomes after transfer. Fertil Steril. 2004 Jul;82(1):108-18. doi: 10.1016/j.fertnstert.2003.12.024.)
Briefly, grading on day 4 embryo was based on [1] the proportion of blastomeres undergoing the compaction process; [2] the morphology of the compacted multicellular mass; [3] the embryo quality on days 2 and 3; and [4] the amount of fragmentation. Embryo quality was graded from 1 to 3, in which the grade of 3 represented good quality, grade 2 intermediate, and grade 1 poor.
Photomicrographs of morulae at different substages with different quality. (A) Grade 3 early morula showing blastomeres starting to “merge” together, but individual cell profile still distinguishable. (B) Grade 3 compact morula embryo characterized by invisible individual cell, but nuclei spread and distinct. Embryo profile is smooth, which makes the embryo appear as one big cell. (C) Grade 3 late morula in which cell boundaries are distinguishable again and cell number increases compared with early morula. Blastomeres at the periphery become spindle shaped and blastulation starts. (D) Grade 2 compact morula embryo showing about 60%–65% blastomeres undergoing compaction with < 20% fragments. (E) Grade 1 compact morula embryo exhibiting an irregular profile, although all blastomeres are undergoing compaction. On the morning of Day 3 of culture, embryos were observed using an inverted microscope and scored according to the Peter cleavage-stage embryo Scoring system.
Depending on the number and quality of embryos, the severity of adenomyosis, the age of patient, as well as the patient’s wishes and the doctor’s preference, single or double embryos were transferred.
- Ethics Board approval is commonly needed for studies evaluating retrospective data. Authors should provide the protocol number of the approval. If no approval has granted, authors should clearly state the reason.
Response: We thank the reviewer for pointing out it. We added it at the end of the manuscript.
Institutional Review Board Statement: The study was conducted in accordance with the Declaration of Helsinki, and was approved by the ethics committee of Beijing Obstetrics and Gynecology Hospital, Capital Medical University (protocol code 2023-KY-011-01).

Reviewer 2 Report
This is an interesting paper which tries to clarify the best strategy in adenomyosis patients concerning the treatment outcome.
Despite the limitations of the paper like employing different endometrial preparations which can impact on the results, it has been found enough interesting to be published.
However it is suggested that this limitation must be pointed out in the discussion
Author Response
This is an interesting paper which tries to clarify the best strategy in adenomyosis patients concerning the treatment outcome.
Despite the limitations of the paper like employing different endometrial preparations which can impact on the results, it has been found enough interesting to be published.
However it is suggested that this limitation must be pointed out in the discussion
Response: We appreciate the reviewer’s suggestion, and thank you for helping to improve the manuscript.
Different endometrial preparation protocols were employed for FET.

Round 2
Reviewer 1 Report
Authors performed all the required revisions according to this reviewer previous comments. According to this reviewer opinion the manuscript merits publication.